# Towards Understanding Distilled Reasoning Models: A Representational Approach

**David D. Baek, Max Tegmark**
Massacusetts Institute of Technology
Cambridge, MA 02139, USA
{dbaek,tegmark}@mit.edu

## Abstract

In this paper, we investigate how model distillation impacts the development of reasoning features in large language models (LLMs). To explore this, we train a crosscoder on Qwen-series models and their fine-tuned variants. Our results suggest that the crosscoder learns features corresponding to various types of reasoning, including self-reflection and computation verification. Moreover, we observe that distilled models contain unique reasoning feature directions, which could be used to steer the model into over-thinking or incisive-thinking mode. In particular, we perform analysis on four specific reasoning categories: (a) self-reflection, (b) deductive reasoning, (c) alternative reasoning, and (d) contrastive reasoning. Finally, we examine the changes in feature geometry resulting from the distillation process and find indications that larger distilled models may develop more structured representations, which correlate with enhanced distillation performance. By providing insights into how distillation modifies the model, our study contributes to enhancing the transparency and reliability of AI systems.

## 1 Introduction

The field of natural language processing has witnessed a rapid development of large language models (LLMs) over the past decade. Early breakthroughs like the Transformer architecture (Vaswani et al., 2017) revolutionized sequence modeling by utilizing self-attention mechanisms (Cheng et al., 2016). This enabled training on unprecedented scales, leading to the era of foundation models (Kaplan et al., 2020; Hoffmann et al., 2022). OpenAI's GPT series (Achiam et al., 2023) further scaled model sizes and datasets, showing that performance follows predictable power-law scaling laws in model size and data. These scaling efforts often yielded emergent abilities – qualitative leaps in capability not seen in smaller models. Another key breakthrough that has further enhanced the potential of these models is the incorporation of chain-of-thought (CoT) reasoning (Wei et al., 2022). By encouraging models to articulate intermediate reasoning steps, chain-of-thought methods have not only improved task performance but have also enabled more complex, multi-step problem solving.

While most LLM improvements have come from scale and supervised fine-tuning, reinforcement learning (RL) has recently emerged as a promising avenue to instill better reasoning abilities. These approaches have culminated in the development of highly competent reasoning models, such as o1 (Jaech et al., 2024) and Deepseek-R1 (Guo et al., 2025), which exhibit exceptional performance on tasks that demand rigorous logical inference. Through RL fine-tuning, these models learned how to refine its reasoning strategies – recognizing mistakes, breaking down complex problems, and trying alternative approaches. Moreover, the output from these reasoning models has also been used to empower smaller models, the process known as *model distillation* (Polino et al., 2018).

Despite the empirical success of model distillation, a critical gap remains in our understanding of how distillation modifies the model. Therefore, we take the first modest step toward understanding *how* distillation changes the model. Specifically, we aim to address the following three questions:

**Q1:** What distinctive features do distilled models develop, and how do these features relate to the models' reasoning capabilities?

**Q2:** Do distilled models exhibit a greater number of unique features as the base model size increases? If so, how does this divergence scale with model size?

**Q3:** How does the feature geometry change as a result of distillation? Are there indications of more structured or organized representations in distilled models compared to their base counterparts?

By understanding the unique features and changes in feature geometry introduced during distillation, we can gain deeper insights into how distillation modifies the model. This contributes to improving the transparency and reliability of AI systems and provides valuable insights into building safe and robust models.

The remainder of this paper is organized as follows. In Section 2, we review the related literature. Section 3 provides an overview of the sparse autoencoder and the sparse crosscoder. Next, Section 4 examines the unique features of distilled models. In Section 5, we delve deeper into four specific types of reasoning features, and analyze the faithfulness of features via ablation experiments and steering. Section 6 explores the changes in feature geometry resulting from distillation. Section 7 concludes the paper.

## 2 RELATED WORKS

**Mechanistic Interpretability** is a line of research that attempts to reverse-engineer neural networks, by breaking down their computations into human-understandable components. One approach focuses on finding circuits, a sets of neurons and weights that together implement a sub-function in the model (Michaud et al., 2024; Olah et al., 2020; Templeton et al., 2024). A prominent example is the discovery of induction heads (Olsson et al., 2022) – pairs of attention heads that enable a form of copying mechanism crucial for in-context learning. Another approach focuses on studying the representations of neural networks (Liu et al., 2022; Baek et al., 2025; Park et al., 2024; Zhong et al., 2024; Baek et al., 2024). For instance, Kantamneni & Tegmark (2025) found that LLMs use helical representations of numbers to perform addition. Beyond understanding how LLMs operate, mechanistic interpretability can also highlight potential failure modes and suggest ways to mitigate them.

**LLM Representations** Understanding LLM representations is a crucial component of comprehending model behavior. This line of research builds upon the Linear Representation Hypothesis (LRH) (Olah et al., 2020), which posits that each feature corresponds to a one-dimensional direction. Empirical studies have demonstrated that LLMs form linear representations across various domains, including space-time (Gurnee & Tegmark, 2023; Li et al., 2021) and truth values (Marks & Tegmark, 2023). LRH has led to approaches like sparse autoencoders (Lieberum et al., 2024) and transcoders (Paulo et al., 2025) to find interpretable linear combinations of neurons. Some recent works have pointed out potential exceptions to the LRH by revealing multi-dimensional circular features (Engels et al., 2024).

**Model Distillation** is a method for compressing deep neural networks by transferring knowledge from a large, high-performing teacher model to a smaller, more efficient student model. Originally introduced by Hinton (2015), knowledge distillation has now become a pivotal technique for transferring advanced capabilities from large language models (LLMs) to relatively smaller language models (Xu et al., 2024). The success of distillation is exemplified by DeepSeek-R1-Distill-Qwen-32B (Guo et al., 2025), a 32-billion-parameter distilled model that outperforms OpenAI-o1-mini across various benchmarks.

**Model Diffing** refers to an interpretability technique for comparing neural networks. Shah et al. (2023) proposed a framework for comparing two learned algorithms, by finding an input transformation that leaves one output invariant but not the other. Recent work by Lindsey et al. (2024) demonstrated that a sparse crosscoder could be used to compare two models and identify which features are newly introduced by instruction fine-tuning.

**Model Steering** refers to a general method of controlling and modifying models' outputs without additional training. One of the well-known steering methods is activation addition (Turner et al., 2023; Jorgensen et al., 2023; van der Weij et al., 2024), where the model could be steered into behave in a certain way by adding a single feature vector to the activation, for instance, love to

hate direction. Sakarvadia et al. (2023) showed that it is possible to improve factual accuracy on benchmarks by injecting supplemental information via steering.

## 3 PRELIMINARIES

### 3.1 SPARSE AUTOENCODER

Sparse Autoencoder (SAE) is a technique for decomposing model activations into a sparse set of linear feature directions. The forward pass of standard SAEs is defined as

$$SAE(x) = W_{dec}\text{ReLU}(W_{enc}x), \tag{1}$$

where $W_{dec}$ is the decoder matrix, and $W_{enc}$ is the encoder matrix. SAEs are trained to minimize the reconstruction loss, as well as the sparsity loss on activated features:

$$\mathcal{L} = \|SAE(x) - x\|^2 + \text{Sparsity Loss}, \tag{2}$$

where various forms of sparsity loss have been proposed in the literature, including weighted L1 (Bricken et al., 2023; Cunningham et al., 2023), JumpReLU (Rajamanoharan et al., 2024), and TopK (Gao et al., 2024).

### 3.2 SPARSE CROSSCODER

Sparse Crosscoder (Lindsey et al., 2024) is a variant of SAEs that allow examining interactions between different activations, for example, from different models, different layers, or different context positions. To train a sparse crosscoder using activations from two models $A$ and $B$, the crosscoder feature activation is computed as

$$f(x_j) = \text{ReLU}\left(\sum_{i=A,B} W_{enc}^{(i)} a^{(i)}(x_j) + b_{enc}\right), \tag{3}$$

$$a'^{(i)}(x_j) = W_{dec}^{(i)} f(x_j) + b_{dec}^{(i)}, \tag{4}$$

where $a^{(i)}(x_j)$ is the activation of model $i$ at token $x_j$, and $a'^{(i)}(x_j)$ is the reconstructed activation.

The crosscoder is trained to minimize the loss, which is a sum of reconstruction MSE and the sum of per-feature decoder vector's L2 norm:

$$\mathcal{L} = \sum_{i=A,B} \|a'^{(i)} - a^{(i)}\|^2 + \sum_k f_k(x_j) \sum_{i=A,B} \|W_{dec,k}^{(i)}\|. \tag{5}$$

## 4 UNIQUE FEATURES OF DISTILLED MODELS

We train a sparse crosscoder with 32768 features on 200 million tokens from the huggingface dataset `open-thoughts/OpenThoughts-114k` and another 200 million tokens from the huggingface dataset `togethercomputer/RedPajama-Data-1T-Sample`. The former includes math, science, and code reasoning traces generated by DeepSeek-R1, whereas the latter includes general text dataset; In this way, we aimed to identify both reasoning and general text features of the models. We analyze three distilled models: DeepSeek-R1-Distill-Qwen-1.5B, 7B and 14B. The crosscoder was trained to reconstruct a residual stream input to the half depth layer of each model. To identify features unique to each model, we define the relative decoder norm as the ratio between the L1 norm of the decoder vector for each model:

$$\text{Relative Decoder Norm (RDN)} = \frac{\|W_{dec,k}^{(B)}\|_1}{\|W_{dec,k}^{(A)}\|_1} \tag{6}$$

$$\text{Normalized Relative Norm (NRN)} = \frac{\text{RDN}}{1 + \text{RDN}} \tag{7}$$

Table 1: Exemplary Features of Qwen-1.5b Crosscoder with Activating Examples.

| Feature | Activating Examples |
|---------|---------------------|
| Self-reflection reasoning | (a) 3. Therefore, the altitude from B is perpendicular to AC, so its slope is the negative reciprocal: 3/4. The altitude passes through B(0,0), so its equation is y = (3/4)x. Find the intersection of x=5 and y=(3/4)x. When x=5, y=15/4=3.75. Therefore, the orthocenter H is at (5, 15/4). Wait, let (b) that's a clue. So the total numbers are n*(n+1). Let's check for n=1. Then it would be 1*2=2 numbers. But the pattern for n=1 would be a single line. Let's imagine: perhaps for n=1, the line is 1*2. But maybe the problem's examples start from n=2. But the constraints say $1 \le n \le 70$, so I need to handle all cases. But let |
| Computation Verification | (a) YES. Test case 2: 00110011. The string is 0,0,1,1,0,0,1,1. $First_1$ is 2, $last_1$ is 7. The substring from 2 to 7+1 (8) is S[2:8], which is '110011'. There are '0's here. So the check fails. Hence, output NO. Test case (b) check again: Wait, first term from a $2^2 * (a + b + c) : a(a + b + c)x^2$ Second term from $bx * (2a + b)x : b(2a + b)x^2$ Third term from $c * ax^2: acx^2$ So combining $x^2$ terms: $a(a + b + c) + b(2a + b) + ac = a^2 + ab + ac+$ |

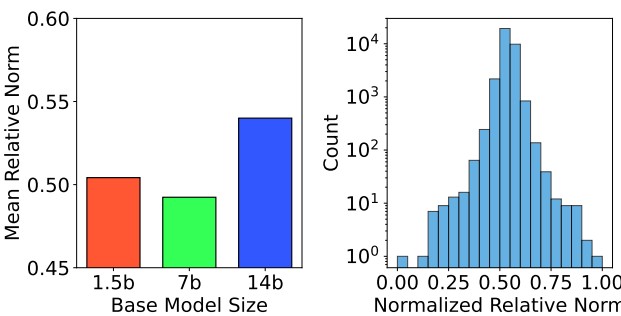

Figure 1: (Left) Average normalized relative norm across all features for base models of various sizes. (Right) Distribution of normalized relative norm for Qwen-14b crosscoder features.

In our discussion, we set model $A$ as the base model, and $B$ as the reasoning model. In this case, shared features correspond to RDN = 1, and NRN = 0.5. RDN $\to 0$ and NRN $\to 0$, correspond to features that are unique to base model. RDN $\to \infty$ and NRN $\to 1$, correspond to features that are unique to the distilled model.

We first examine the distribution of relative decoder norms. Figure 1 shows the distribution of normalized relative norm, as well as the average NRN across models of different size. We observe that most features are indeed shared features (NRN = 0.5), with exponentially decaying number of features on each tail of the distribution. Average relative norm is particularly larger for Qwen-14b crosscoder, indicating that larger distilled models may have more unique features than the smaller models.

Upon sorting features by NRN, we find the features unique to distilled model, as shown in Table 1. We asked GPT-4o-mini to annotate top 100 features and bottom 100 features. Unique features of distilled models include various reasoning features, such as (a) *self-reflection reasoning*, where the model recognizes that its thinking process may be incorrect and corrects itself, often using the word *wait*; and (b) *Verification reasoning*, where the model verifies that its solution is indeed a correct answer to the question. We found that bottom 100 features sometimes activate on reasoning contexts

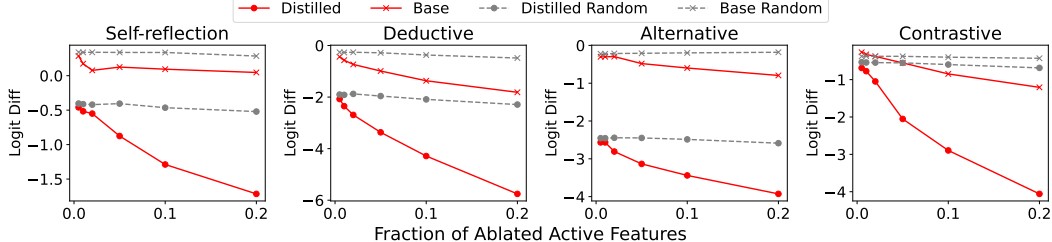

Figure 2: Ablation Experiment: Histogram depicting the average logit change in both base and distilled models as a result of ablating features with (a) NRN $> 0.5$, and (b) firing frequency in top $k \in [0.5, 1, 2, 5, 10, 20]\%$.

as well; however, we believe that base model's crosscoder feature simply learns the token feature *wait*, since (a) we were not able to steer the base model into self-correcting more by manipulating the relevant latents; (b) feature ablation experiment suggests that these reasoning features may likely be acausal in the base model; and (c) base model does not self-correct that much and the average logit of words such as 'wait' is significantly smaller than the distilled model, so it is unlikely for a base model to develop a self-reflection reasoning feature. We present the results of steering experiment and ablation experiment in the next section.

## 5 CASE STUDY: REASONING FEATURES

In this section, we study four specific types of reasoning features: (a) self-reflection, (b) deductive reasoning, (c) alternative reasoning, and (d) contrastive reasoning. We examine how these reasoning features are distributed along the normalized relative decoder norm (NRN) dimension. To identify each reasoning context, we collect activations from the following target tokens:

- Self-reflection: "Wait,"
- Deductive: "Therefore," "Thus"
- Alternative: "Alternatively"
- Contrastive: "But," "However"

Here, we hypothesized that each reasoning feature will likely fire frequently on related target tokens. In order to testify the causality of these reasoning features, we perform ablation experiments. Among all active features firing on the above target tokens, we zero-ablate features with (a) NRN $> 0.5$, and (b) firing frequency in top $k \in [0.5, 1, 2, 5, 10, 20]\%$ in each reasoning category. We then measure the average logit change across 100 randomly chosen target tokens. Figure 2 shows the results of our ablation experiments. We find that across all reasoning categories, the distilled model's target logit drops significantly as a result of ablation, while the base model's target logit tends to remain identical. This experiment verifies that a set of reasoning features are causally responsible for performing certain types of reasoning in distilled models, and that such causal effects are relatively small for base models.

We also found that certain features may be highly relevant in determining how long the model thinks. For example, distilled model's behavior could be steered into 'over-thinking' mode or 'incisive-thinking' mode, as shown in Figure 3 and Figure 9. When we steer the model by adding the feature's decoder vector (single feature) to the model, the model enters the 'over-thinking' mode, where the model thinks for extremely long, even for a very simple question like 'What is 5-1?'. Moreover, the model checks their answer multiple times until they stop thinking and conclude their answer is correct.

Distilled model could also be steered into 'incisive-thinking' mode. We hypothesize that such incisive-thinking feature may be highly correlated with the deductive feature defined above. Deductive feature fires around the concluding statement, and therefore we believe steering the model via deductive feature's decoder vector (single feature) may make the model more confident. In

**Distilled Model: Before Steering**

What is the next number in the sequence: 16, 32, 48, ...? <think>

Okay, so I have this sequence here: 16, 32, 48, and I need to figure out what the next number is. Hmm, let me think about how sequences work. They often follow a pattern, whether it's addition, multiplication, or some other operation.

First, let me look at the differences between the numbers. The first number is 16, the second is 32, so the difference between them is 16. Then, the second to the third is 32 to 48, which is also 16. So, the difference is consistent at 16. That suggests that the sequence might be increasing by 16 each time. If that's the case, the next number would be 48 plus 16, which is 64.

But wait, maybe that's too simplistic. Maybe it's not a straightforward addition. Sometimes sequences involve multiplication or exponential growth. Let me check if multiplying by a constant gives the right result. If I multiply 16 by 2, I get 32, which matches the next number. Then, multiplying 32 by 1.5 gives 48, which is correct. But if I multiply 48 by, say, 1.25, I get 60, which is different from 64.

Hmm, another approach is to see if it's a linear relationship. If I let n represent the position of the number (like n=1 for the first number), then the first number is 16, the second is 32, the third is 48, and so on. If I set up the equation, like:

$a_n = a_1 + (n - 1)d$

where d is the common difference. Earlier, I found that d is 16, so substituting, I get:

$a_n = 16 + (n - 1) * 16$  $a_n = 16(1 + n - 1)$

[Dividing into $16 * (1 + n - 1)$] $a_n = 16n$

So, for n=1: 16*1=16, n=2:16*2=32, n=3:48, which checks out. So, the general term is 16n, which makes the next term at n=4 equal to 64.

But why did I have this uncertainty earlier about multiplication? Because in the first two numbers, 16 and 32, 16 multiplied by 2 gives 32, but then 32 multiplied by 1.5 gives 48, which is different from just 48 times something else. However, if I use the linear term, it's still 64.

··· (Continued)

**Distilled Model: After Steering**

What is the next number in the sequence: 16, 32, 48, ...? <think>

First, I'll analyze the given sequence: 16, 32, 48, and determine the differences between consecutive terms to identify any pattern.

Looking at the differences: - 32 - 16 = 16 - 48 - 32 = 16

It appears that the sequence increases by 16 each time. To confirm this pattern, I can test the next term by adding another 16 to the last term:

48 + 16 = 64

Thus, the next number in the sequence would be 64. < /think>

To determine the next number in the sequence **16, 32, 48, ...**, let's analyze the differences between consecutive terms.

1. **Find the differences:**

- $32 - 16 = 16$ - $48 - 32 = 16$

It appears that each term in the sequence increases by **16**.

2. **Continuing the pattern:**

- Next term: $48 + 16 = 64$

**Final Answer:**

$\boxed{64}$

Figure 3: Distilled Model's behavior steered into incisive thinking mode.

particular, original distilled models tend to repeat or check the same statement multiple times, especially when the problem becomes complex. Steered model, however, tends to be more confident and incisive in their thinking process.

## 6 FEATURE GEOMETRY OF DISTILLED MODELS

Does the distilled model have better feature geometry? One possible explanation for improved LLM capabilities is that these models build better representations – much like how humans enhance their understanding of a subject by *connecting the dots* and organizing their knowledge more effectively. To investigate whether distilled models have more structured representations, we measure the quality

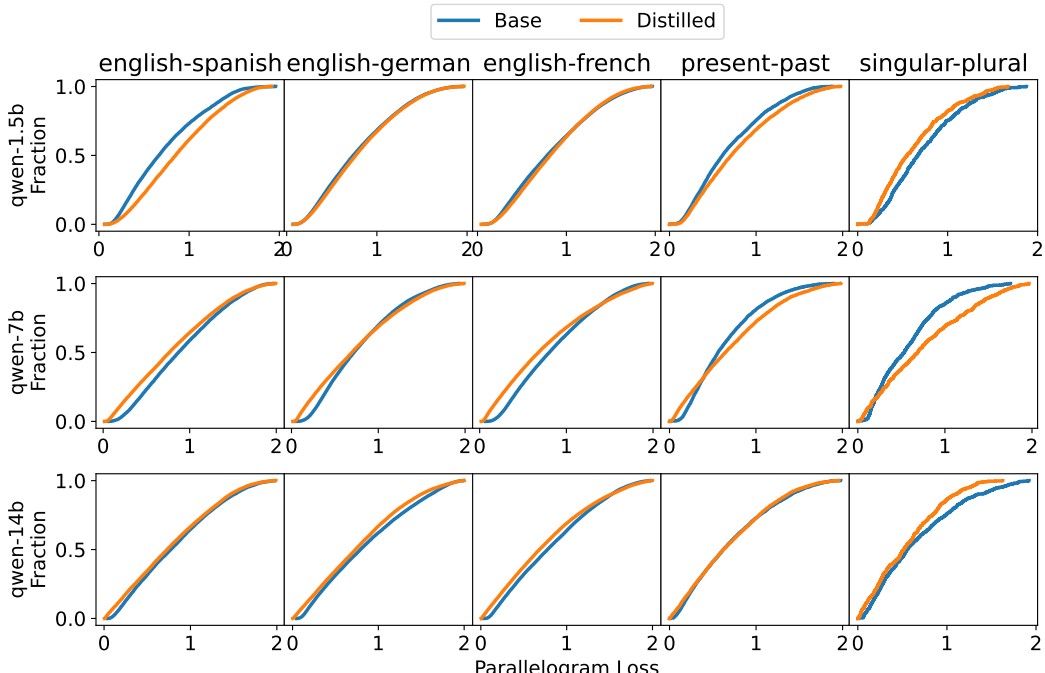

Figure 4: Cumulative fraction as a function of parallelogram loss for different models and function classes. Distilled model's representations tend to become more structured as the model scales.

of semantic parallelograms (e.g., the classic example: man:woman::king:queen). We use the dataset from Todd et al. (2023). This dataset consists of a pair of words that are related by a specific function. By using two pairs from the same function class, one can construct a semantic parallelogram. We then measure the parallelogram loss; Let $(\mathbf{E}_a, \mathbf{E}_b, \mathbf{E}_c, \mathbf{E}_d)$ be the PCA-ed activations. Then, the parallelogram loss is defined as

$$\text{Parallelogram Loss} = \frac{\|\mathbf{E}_a - \mathbf{E}_b - \mathbf{E}_c + \mathbf{E}_d\|}{\sqrt{\|\mathbf{E}_a\|^2 + \|\mathbf{E}_b\|^2 + \|\mathbf{E}_c\|^2 + \|\mathbf{E}_d\|^2}}. \tag{8}$$

For each function class, we first select entries that consist of a single token, and then compute the parallelogram loss over all possible pairs. We use the residual stream input activations from the half-depth layer of the model, reduce the representations to 20D using PCA, and then evaluate the parallelogram loss. Figure 4 shows the parallelogram loss for different models and function classes. While the base model generally outperforms the distilled model in Qwen-1.5B, we observe that the distilled model starts to outperform the base model as the model size increases. In fact, the 14B distilled model exhibits better parallelogram performance than the base model across all evaluated function classes. This implies that as model size scales, distilled models achieve better-structured representations, which subsequently leads to improved distillation performance. In Appendix A, we also show that such structure improvement is robust against the number of PCA dimensions we choose.

## 7 CONCLUSION

In this paper, we examined how model distillation impacts the development of reasoning features in LLMs. We find various reasoning features from the sparse crosscoder, such as self-reflection and computation verification feature. In particular, we observe that distilled models contain unique reasoning feature directions, which could be used to steer the model into over-thinking or incisive-thinking mode. Lastly, we find indications that larger distilled models may develop more structured representations, which correlate with enhanced distillation performance. Ultimately, our work contributes to improving the transparency and reliability of AI systems by providing insights into how distillation modifies the model.

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

# A   PARALLELOGRAM LOSS FOR DIFFERENT PRINCIPAL COMPONENT DIMENSIONS

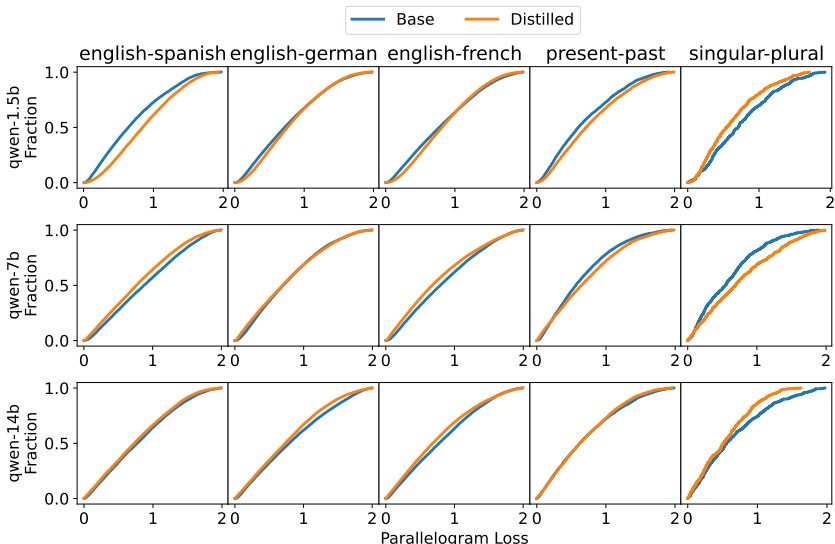

Figure 5: Parallelogram loss with activations PCA-ed into 2D.

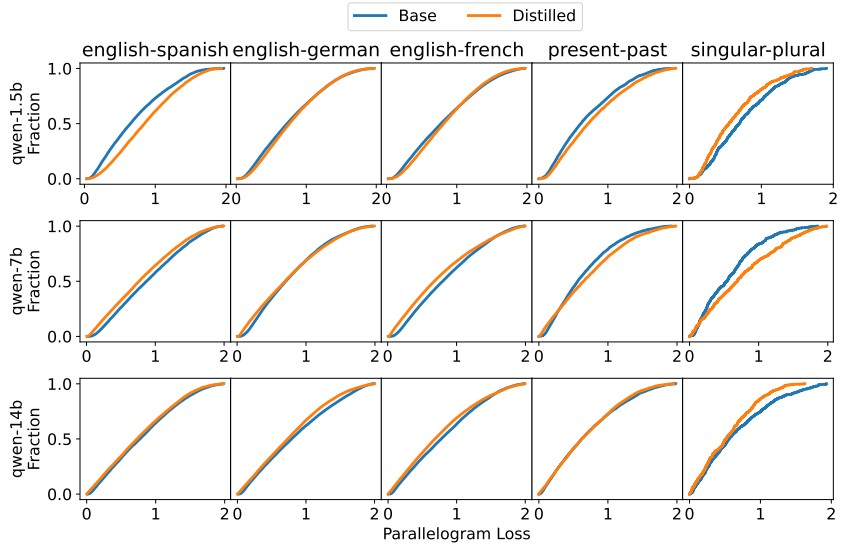

Figure 6: Parallelogram loss with activations PCA-ed into 5D.

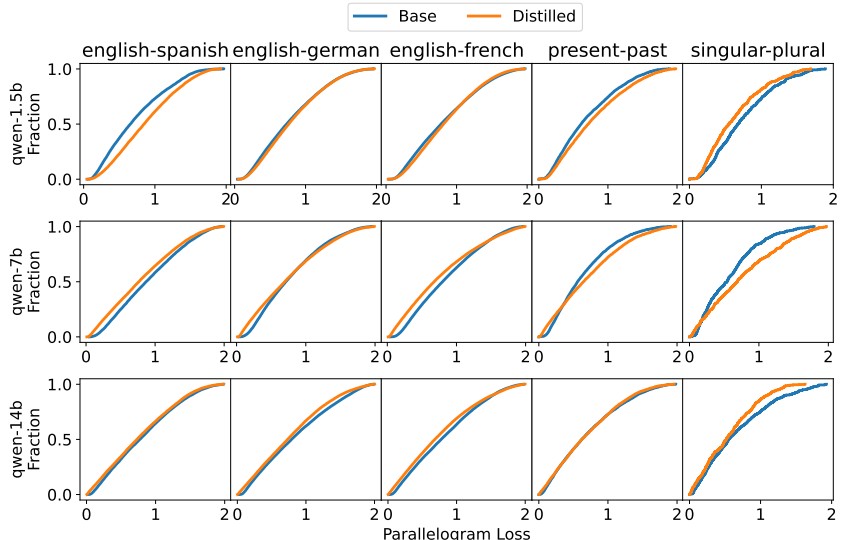

Figure 7: Parallelogram loss with activations PCA-ed into 10D.

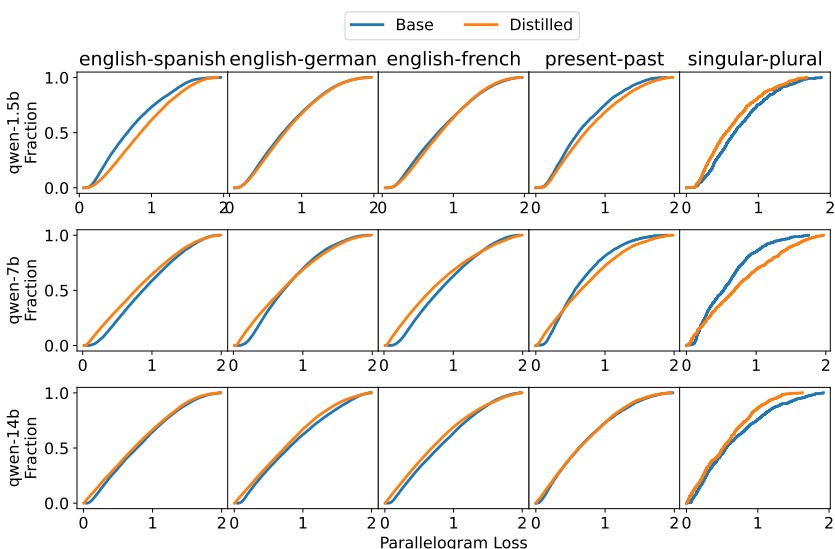

Figure 8: Parallelogram loss with activations PCA-ed into 20D.

## B   EXAMPLE OF STEERED OUTPUTS

**Distilled Model: Before Steering**

What is 5-1? <think>

To determine what 5 minus 1 equals, I start by recognizing the numbers involved. The number 5 is being subtracted by 1, which is a basic arithmetic operation.

Next, I perform the subtraction as indicated. Subtracting 1 from 5 results in 4.

Finally, I confirm that the result is correct by adding the result (4) to the smaller number (1) to see if it equals the original larger number (5). Since 4 plus 1 equals 5, the calculation is correct.

I conclude that 5 minus 1 indeed equals 4. </think>

To find the value of $5 - 1$, follow these steps:

1. **Identify the Numbers Involved:** - Minuend (the number being subtracted from): 5 - Subtrahend (the number being subtracted ): 1

2. **Perform the Subtraction:**

$$5 - 1 = 4$$

3. **Final Answer:**

$$\boxed{4}$$

**Distilled Model: After Steering**

What is 5-1? <think>

Okay, so I need to figure out what 5 minus 1 equals. Hmm, let's start by visualizing this with something I know, like money or maybe blocks. If I have 5 coins and someone gives me 1 coin, how many do I have left? That makes sense.

Alternatively, I could think of it in terms of numbers on a number line. If I start at 5 and jump back one step, where do I land? Let me count down from 5. That would be 4, 3, 2, and finally 1. Wait, so I'm at 1 then. That seems right.

But maybe I can use some bodily actions to double-check. If my fingers are on table and I show five fingers, then I take one away, I should have four fingers left. That gives me another confidence that 5 minus 1 is 4.

Another way is to count backwards. Starting from 5 and counting back one: 4, 3, 2, 1. Yep, that's four. So it seems like whatever method I use, I end up with 1 as the answer after 5 minus 1.

Wait, could there be a mistake here? Did I forget something or oversimplify? Let's think. 5 minus 1 is straightforward. No fractions, no negatives, just a simple subtraction. So yeah, it's definitely 4.

Wait a second, maybe I'm second-guessing myself too much. Did I not use a real number or just a calculation without context? Well, the problem says 5 minus 1, so it's a basic subtraction. No, I think I have it right.

Is this a case where I missed something, like borrowing in subtraction? No, in this case, I don't need to borrow. It's just a basic subtraction without any borrowing involved. So yeah, nothing here to complicate things further.

Perhaps I can also use real-world objects, like apples, to apply this. If I have 5 apples and someone takes 1, how many do I have left? Starting with 5, take 1, what's left? So, 4 apples. That also makes sense.

Wait, if I play with real coins and take one away, 5 minus 1, is that four coins? Yeah, that's the same thing.

But I can also do square roots or something, but that's overcomplicating. Wait, but maybe not. Let me think, if I don't know this subtraction, maybe it's in a different context.

$\cdots$ (Continued)

Figure 9: Distilled Model's behavior steered into over-thinking mode.

