# OpenReview forum: "Towards Understanding Distilled Reasoning Models: A Representational Approach"
_ICLR.cc/2025/Workshop/BuildingTrust — BuildingTrust_

### Official Review · Reviewer_s2qG · 2025-02-22
**Interesting work on reasoning in distilled models**

**Rating:** 6
**Confidence:** 3

**Review:**

This work compares the reasoning behavior across distilled LLMs using crosscoders.
Overall, I'm recommending an acceptance because I think the ideas are interesting, and would make for an interesting discussion at the workshop. Moreover, I think this could be a great conference paper with further work. That being said, the authors need to be more careful in the claims they make and improve the writing of the paper/ experiments. Below, I've elaborated on areas that need improvement.

[1] Claims

The current research questions are not (fully) reflected or supported by the experiments. Below, I've suggested modifications and/or commented on the types of experiments I would have expected to see.

Q1: "What distinctive features do stilled models develop, and how do these features relate to the model reasoning abilities?" -> "What is the overlap between the features represented by the distilled models base model". The authors need to provide further analysis of the features represented if they want to claim that they identify features that differ.

Q2 -> "Do distilled models exhibit a greater number of unique features as the base model size increases? If so, how does this divergence scale with model size?" For the second claim, I'd expect the authors to propose a scaling law, or more generally highlight a trend. However, this is not in the current version of the paper.

Further, please be careful with what you attribute to model behavior. It's often possible that the observed trends are a result of the interpretability tool (crosscoders). More concretely, it is possible that the base model and distilled model encode the same feature, but that it is not identified up by the crosscoder.
The authors have tried to tackle this through an ablation study in Section 5. However, I think this ablation measures faithfulness -- whether the features influence the output -- which is not the same.

[2] Feedback on experiments

Section 5: How are the ablations performed? (What values are the features set to?)
More generally, I'd encourage the authors to be more careful in their conclusions. This experiment seems to test whether the features influence the model output -- not whether the features are unique. To claim that the features are unique, you'd need to show that they do not exist in the other LLM.

Regarding the ablation results for the base model, it might be explained by https://arxiv.org/abs/2307.15771.

Section 6: Feature geometry

The experiment setup and results in Section 6 are unclear -- why is this chosen above the original setup in Todd et al. (2023) (from which the task is taken)? In particular, using the first two principal components heavily influences the results. What percentage of the latent space is explained by the first two PCs? Usually the first two PCs are only chosen for visualisation purposes. I think this needs to be further motivated and analyzed. Currently, I'd strongly recommend using the original setup of Todd et al. (2023) instead.

[3] Writing and References

The authors should be more careful with references/ attributions of prior results:
- Waswani-> Vaswani
- Vaswani et al. did not introduce self-attention, the correct reference is https://arxiv.org/pdf/1601.06733
- https://arxiv.org/pdf/2001.08361 and https://arxiv.org/abs/2203.15556 are the correct references for scaling laws of LLMs

[4] Precision of Language
- Line 039-> RLHF is RL, so I’d be careful
- Line 266: PCAed <- this type of language is too informal; please be careful with this more generally. Projections is more appropriate.
- LRH: “feature corresponds to a one-dimensional direction” -> networks represent features as linear directions. One-dimensional would imply a single neuron.
- Equation 3 is missing a sum before the first part.

---

### Official Review · Reviewer_vDiT · 2025-02-24

**Rating:** 5
**Confidence:** 4

**Review:**

**Paper Summary**

This paper presents an empirical study of the representational similarity between distilled reasoning models and their base counterparts. The authors employ sparse crosscoders trained on pairs of base and distilled Qwen models to quantify the degree to which features are shared between the models. The paper highlights examples of features that appear exclusively in the distilled models and examines the geometry of the representation space, noting that models with larger parameter counts tend to have a more structured representation space.

**Strengths**

- The use of a crosscoder to analyze distilled reasoning models is a creative approach.
- The method for distinguishing shared versus non-shared latents based on the relative decoder norm seems to be a promising direction.

**Weaknesses**

- The paper does not provide sufficient training details for the crosscoder. It is unclear whether the crosscoder achieves a satisfactory reconstruction loss, and details such as the number of dead latents or the sparsity details are not reported.
- In Section 4, the process by which the authors assign meaning to “reasoning” latents is not clear. The paper describes three latents from the top 16 sorted by normalized relative norm and provides descriptive labels along with an “activating example” for each. However, it is not clear how these descriptions were derived. How was the description obtained? How was the activating examples chosen, is the max-activating example? Are other activating examples consistent with the expiation provided fro the latent?
- The takeaways from Figures 2 and 3 are not clearly stated. Can we conclude anything more than “most features are shared between the base and distilled models”?
- Some additional details about the experiment in section 6 would be appreciated. E.g., how are the PCA-ed activations computed? What is an example of a semantic parallelogram form the dataset?
- Nothing is said about the norm of the encoder. For instance, are there cases of latents for which the normalized relative norm of the decoder is 1 or 0, but the same metric for the encoder is around 0.5, or vice-versa?

**Conclusion**

Overall, crosscoder-based approach for analyzing distilled reasoning models is interesting, but the experimental rigor and clarity could be improved.

---

### Official Review · Reviewer_MB4z · 2025-03-01
**Review of “Understanding Distilled Reasoning Models: A Representational Approach”**

**Rating:** 5
**Confidence:** 2

**Review:**

### Summary
This paper studies the effect of model distillation on reasoning features in Large Language Models (LLMs). By using a sparse crosscoder approach, the authors compare activation patterns in base vs. distilled Qwen-series models, highlighting “unique” directions in which the distilled models appear to form or reshape reasoning features (e.g., self-correction, heuristic reasoning). The analyses suggest that larger distilled models may form more structured representations, as evidenced by lower parallelogram losses on select tasks.

---

### Strengths
- **Novel angle on distillation**: The idea of examining new “reasoning directions” induced by distillation helps clarify how knowledge transfer might alter internal representations.
- **Sparse crosscoder technique**: Using this approach to isolate features linked to reasoning tokens (“wait,” “therefore,” etc.) is a creative application of interpretability methods.
- **Structured Representations**: The parallelogram-loss analysis for semantic relationships is interesting, and the results that bigger distilled models might achieve better “feature geometry” is a notable hypothesis for future work.

---

### Weaknesses and Concerns
1. **Limited scope and generalizability**: The paper only evaluates Qwen-based models. It’s unclear whether these results translate to other architectures or instruction-tuned variants.
2. **Focus on a single interpretability method**: While the sparse crosscoder is valuable, readers may wonder whether additional methods (e.g., neuron-level causal interventions) could corroborate the findings.
3. **Distillation vs. other fine-tuning effects**: The paper does not fully separate how much of the “unique” directions in reasoning come specifically from distillation (rather than dataset differences, chain-of-thought, or RLHF).
4. **Methodological clarity**: Certain details, like how threshold values were chosen or the exact distribution of data from which tokens were sampled, could be more thoroughly described for easier reproducibility.
5. **Lack of detailed analyses**: More analyses or experiments to interpret the results more clearly would make the results much more compelling.

---

### Recommendation Rationale
Despite the interesting premise, the current results feel somewhat narrow in scope, leaving questions about broader applicability and the causal mechanism driving the emergent reasoning directions. Additional experiments (e.g., ablations or controls with different base architectures) would strengthen the paper. As it stands, I consider it **marginally below** the acceptance threshold due to limited robustness of evidence and scope.

Nevertheless, the direction is promising, and further exploration into whether these findings hold across tasks and model families could produce valuable interpretability insights. With more rigorous experiments and a broader set of models, this work could contribute substantially to our understanding of how knowledge distillation shapes model reasoning.

---

### Decision · Program_Chairs · 2025-03-05

**Decision:**

Accept

**Comment:**

This paper presents an interesting approach to analyzing reasoning features in distilled LLMs using a sparse crosscoder framework. There are novel insights into how distillation impacts model reasoning representations. The paper highlightsdifferences in reasoning feature geometry between base and distilled models. All reviewers acknowledge the value of the research direction but point out limited scope. Despite these limitations, the work presents an innovative framework that warrants discussion at the workshop. Given the interesting ideas and potential for further exploration, I recommend acceptance as a workshop paper.